# Preparation and Characterization of Polyphenol-Functionalized Chitooligosaccharide Pyridinium Salts with Antioxidant Activity

**DOI:** 10.3390/md23040150

**Published:** 2025-03-30

**Authors:** Zhen Qi, Wenqiang Tan, Zhanyong Guo, Aili Jiang

**Affiliations:** 1College of Life Sciences, Yantai University, Yantai 264005, China; qzqq2022@163.com; 2School of Marine Science and Fisheries, Jiangsu Ocean University, Lianyungang 222005, China; wqtan@jou.edu.cn

**Keywords:** chitooligosaccharide, phenolic acids, antioxidant activity

## Abstract

As a kind of eco-friendly material with wide application prospects, chitooligosaccharide (COS) has attracted increasing attention because of its unique bioactivities. In this study, novel polyphenol-functionalized COS pyridinium salts were designed and synthesized. The structural characteristics of the desired derivatives were confirmed by FT-IR and ^1^H NMR spectroscopy. Their antioxidant activities were evaluated in vitro by DPPH radical scavenging assay, superoxide anion radical scavenging assay, and reducing power assay. The solubility assay in common solvents and cytotoxicity assay against L929 cells using the MTT method in vitro were also performed. The antioxidant assay results showed that the compounds functionalized by polyphenol displayed improved antioxidant activities, which were enhanced with the increase of sample concentration and the number of phenolic hydroxyl groups. The solubility assay indicated that the prepared derivatives had good water solubility. Besides, the modified products were non-toxic to the cells tested. In short, the polyphenol-functionalized COS pyridinium salts with enhanced antioxidant activity and good biocompatibility could be employed as newly safe antioxidant in the fields of biomedicine and food.

## 1. Introduction

Reactive oxygen species (ROS) are by-products generated by normal cellular energy metabolism, including hydroxyl radicals (·OH), superoxide anion radicals (·O^2−^), hydrogen peroxide (H_2_O_2_), singlet oxygen (^1^O_2_), and so forth, participating in many physiological processes in living organisms, such as cell signaling and immune response [1,2,3]. Normally, the production and removal of ROS is maintained in a state of dynamic equilibrium to keep the regular function of cells [4,5,6]. However, when the production of ROS is excessive or the antioxidant defense system is damaged, superfluous ROS will lead to oxidative stress, and its hyperreactivity could cause harm to biological macromolecules, such as proteins and DNA, thereby inducing the occurrence of chronic diseases including cancer and diabetes [2,7,8,9]. In recent years, exogenous antioxidants have attracted a great deal of attention due to their excellent effects on neutralizing and scavenging oxygen radicals to protect cells from oxidative stress [10]. However, antioxidants of entirely synthetic origin, such as butylated hydroxylanisole (BHA) and butylated hydroxyltoluene (BHT), are being questioned due to some potential side effects on the human body, including certain toxic effects and carcinogenesis at high doses [11,12,13]. Therefore, it is necessary to develop novel safer antioxidants prepared from biomaterials of natural origin for use in areas closely related to human health, such as pharmaceuticals and food industries.

Polyphenols, the most common secondary metabolites derived from plants, are characterized by at least one aromatic ring along with one or more hydroxyl moieties [14]. Phenolic acids, as a significant subgroup of phenolic compounds, are known for their diverse biological activities, including antioxidant, antimicrobial, and anti-diabetic [15,16]. Most of these biological activities are largely dependent on the interaction of phenolic acids with ROS [17,18]. Hydroxybenzoic acids and hydroxycinnamic acids are the two main categories of phenolic acids. Among them, protocatechuic acid and gallic acid belong to the former, distinguished by the presence of carboxyl groups attached to the aromatic ring [19]. This structural feature endows them with significant antioxidant activity, enabling their extensive application as antimicrobial agents [20,21], dietary supplements [22], food preservatives [23], and active food packaging materials [24] in the food and pharmaceutical industries.

Chitosan (CS), the only natural cationic polymer in nature, is primarily derived from chitin sourced from marine organisms through enzymatic hydrolysis, chemical treatment, and microbial fermentation [25]. Its unique coordination structure and cationic properties endow it with a variety of excellent physicochemical properties and biological activities [26]. However, its poor water solubility severely hinders its further development [9]. Its degradation product, chitooligosaccharide (COS), makes up for the shortcoming of poor water solubility of CS and is endowed with more biological activities, including anti-tumor, anti-inflammatory, and antibacterial, which significantly expands the application fields of CS [27]. CS and COS could be further chemically modified to enhance their bioactivities by leveraging the active moiety (amino and hydroxyl groups) on their backbones [28]. For instance, grafting polyphenols onto CS/COS via enzyme or chemical reagent-mediated ways not only combines the advantages of both components but also imparts novel properties to the resulting materials, often exhibiting amplified bioactivities compared to the corresponding polyphenols [29,30].

Over the last decades, there have been numerous studies on the grafting of polyphenols onto CS/COS to obtain modified materials [30]. Xie et al. prepared the copolymer gallic acid-grafted-CS using a developed one-pot method with EDC and HOBt, and the study indicated that the conjugate exhibited stronger antioxidant activity than free gallic acid and a higher viscosity than plain CS [31]. Mittal et al. synthesized five kinds of COS-polyphenol conjugates using gallic, caffeic, and ferulic acids, epigallocatechin gallate, and catechin through the AsA/H_2_O_2_ redox pair hydrolysis method. The results showed that COS-catechin had the highest antioxidant, antidiabetic, and antimicrobial activities than COS and all other COS-polyphenol conjugates under the established experimental conditions [32]. Singh et al. used epigallocatechin-3-gallate (EGCG) for the preparation of COS-EGCG conjugates by a free radical grafting method and the study found that the production had higher antioxidant and antimicrobial activities than COS [33]. According to the above, the polyphenol-CS/COS conjugates, with excellent antioxidant and antibacterial activities, can serve as natural bioactive materials for the development of products with similar functionalities. For example, they can be used to develop active food packaging films, functional food additives, bioactive medical dressings, and so on. In summary, the conjugates prepared by graft copolymerization of CS/COS and polyphenols have achieved the targeted construction of various biomaterial composites, significantly optimizing the desirable properties of individual components [30]. Considering that the interaction between antioxidants and ROS typically occurs in aqueous environments, ensuring the water solubility of antioxidants is of vital importance [31]. To our knowledge, no study has reported the preparation of polyphenol-functionalized COS pyridinium salts for the evaluation of antioxidant activity.

In the present study, the novel polyphenol-functionalized COS pyridinium salts were synthesized via niacylation, salt-forming reaction, and chemical coupling methods. The chemical structures of the derivatives were characterized by FT-IR and ^1^H NMR. In addition, DPPH radical scavenging assay, superoxide radical scavenging assay, and reducing power assay were conducted to estimate the antioxidant capacity of novel prepared derivatives. The solubility of synthesized derivatives in several common solvents was simply performed. Furthermore, the cytotoxicity experiment of new polyphenol-functionalized COS pyridinium salts against L929 cells using the MTT method in vitro was testified.

## 2. Results

### 2.1. Chemical Synthesis

Polyphenol-functionalized COS pyridinium salts were prepared in three steps (Figure 1) in our work. The first step was to obtain nicotinated chitooligosaccharide (NC) using CDI as a catalyst to activate nicotinic acid. Then aminoethyl nicotinated chitooligosaccharide (ANC) was synthesized by nucleophilic substitution reaction. Lastly, benzoic acid, protocatechuic acid, and gallic acid were respectively bound to ANC by amide bonds, resulting in the preparation of benzoic acid, protocatechuic acid, and gallic acid-functionalized chitooligosaccharide pyridinium salts (BANC, PANC, and GANC). Among them, BANC was used as the control group due to the absence of phenolic hydroxyl groups on its benzene ring structure.

### 2.2. Characterization

#### 2.2.1. FT–IR Spectra

The chemical structures of COS, NC, ANC, BANC, PANC, and GANC were confirmed by FT-IR spectra and shown in Figure 1. The characteristic absorption bands of COS were shown as follows: a broadband appearing at 3394 cm^−1^ is assigned to the N-H and O-H stretching vibration, the absorption peak of 2931 cm^−1^ is attributed to the vibration of C-H, and the bands formed at 1583 cm^−1^ and 1088 cm^−1^ are respectively ascribed to the bending vibration of N–H and the stretching vibration of C–O–C on the COS skeleton [34]. For NC, after nicotinylation reaction, two sets of new absorption peaks appeared at 3075 cm^−1^, and 741–702 cm^−1^ might be related to the vibrational absorption of C–H and C=C bonds on the pyridine ring, severally [35]. In addition, a series of significant strong peaks at 1729 cm^−1^, 1650 cm^−1^, and 1284 cm^−1^ could be concluded to be ester bond (C=O), amide bond (C=O), and ester bond (C–O), which demonstrated the successful introduction of nicotinic acid onto the COS backbone [36]. After nucleophilic substitution reaction, a new peak at 1636 cm^−1^ is observed, which is associated with N–H on aminoethyl [37]. Additionally, the previous C-H vibrational peak at 3075 cm^−1^ is red-shifted to 3065 cm^−1^ due to the introduction of electron-withdrawing group NH_2_. As for the three end products BANC, PANC, and GANC, a significant characteristic absorption band at 1593 cm^−1^ is attributed to the C=C stretching vibration, suggesting the generation of a benzene ring successfully [37]. In conclusion, the above data preliminarily proved the successful synthesis of polyphenol-functionalized COS pyridinium salts.

#### 2.2.2. ^1^H NMR Spectra

To further confirm the chemical structures of polyphenol-functionalized COS pyridinium salts specifically, ^1^H NMR spectra were used to elucidate the changes within functional groups. These changes, crucial for the identification and assignment of molecular structures, are depicted in Figure 2. The chemical shift at 4.79 ppm of the products could be attributed to the D_2_O solvent. For the pure COS, a set of the chemical shifts located at 2.98 ppm, 3.10–4.20 ppm, and 5.30 ppm were derived from the hydrogen protons of H2, [H3–H6], and H1, respectively [38]. As for NC, the modified product bearing the pyridine functional group after nicotinylation reaction, a group of characteristic absorption signals observed at 7.60–9.80 ppm, are attributed to the hydrogen atoms on the pyridine ring, which are marked at the low field of the ^1^H NMR spectra [39]. After introducing the aminoethyl through salt formation, compared to the graph of the NC, the characteristic shifts appeared at 3.70 ppm and 5.04 ppm, and are ascribed to the protons from the two methylene groups on aminoethyl, respectively [40]. Also, compared with ANC, a cluster of the hydrogen proton signals appeared at 6.48–10.00 ppm, and were assigned to the introduction of the benzene/aromatic ring. Overall, these data were enough to demonstrate the successful synthesis of the desired compounds by ^1^H NMR spectra.

### 2.3. Degrees of Substitution (DS) Analysis

The DS of polyphenol-functionalized COS pyridinium salts was calculated by using the integration of [H2–H6] as an integral standard peak in accordance with the ^1^H NMR spectra of COS and its derivatives [41]. As shown in Table 1, NC was obtained by COS directly linked to nicotinic acid with a DS of 1.80 through the catalytic effect of CDI [40]. After the nucleophilic reaction to NC, the DS of ANC was measured to be 0.97. The DS of the three end products derived by COS were as follows: 0.27, 0.30, and 0.30, respectively. Obviously, CDI has a strong catalytic ability for ester and amide bonds, resulting in the highest grafting rate of niacin and COS among the compounds. Moreover, the DS of the three COS derivatives were pretty similar to each other, and the influence of the DS on the experimental results could be ruled out.

### 2.4. Solubility Assay

All sample solutions at a concentration of 10 mg/mL presented in the test tube were shown in Figure 3. From the picture, we were able to see that polyphenol-functionalized COS pyridinium salts demonstrated good water solubility and remained transparent, which was extremely beneficial to the antioxidant effect that occurred in the aqueous environment. The dissolution and precipitation effects of all samples in some familiar reagents were clearly observable and presented in Table 2. “√” meant dissolution, “×” meant precipitation.

### 2.5. Antioxidant Activity

ROS mainly generated by the metabolism of living organisms participate in various biochemical reactions of the human body. In most cases though, the continuous persistence of excess ROS could be closely related to many chronic diseases, which might harm people’s health. The use of antioxidants could be an effective way to deal with this damage caused by reactive oxygen radicals [42]. In our work, the antioxidant activity of all samples was evaluated by DPPH radical scavenging assay, superoxide anion radical scavenging assay, as well as reducing power assay, and the corresponding data were shown in Figure 4, Figure 5, and Figure 6, respectively.

#### 2.5.1. The Scavenging Ability of the DPPH Radical

DPPH (1,1-diphenyl-2-trinitrophenylhydrazine) is a stable free radical with a dark purple color, and the stable solution formed after being dissolved in ethanol can be quantitatively detected by UV-vis spectrometer at 517 nm [43]. When added to the solution, the antioxidants with free radical scavenging capacity could deliver hydrogen atoms or electrons to the lone pair of electrons on the *N* atom attached to the trinitrobenzene ring on the chemical structure of DPPH to form a more stable DPPH-H compound with the decolorization from purple to yellow [44]. The effect of polyphenol-functionalized COS pyridinium salts on scavenging DPPH radicals at different concentrations is shown in Figure 4. From the graph, several conclusions could be obtained as follows. Firstly, the scavenging activity of all samples against DPPH free radicals showed a concentration-dependent manner. With the concentration increasing from 0.10 to 1.60 mg/mL, the DPPH scavenging indexes of all samples were found to be enhanced gradually. Secondly, compared with BANC without phenolic hydroxyl groups as the control, the scavenging rates of PANC and GANC against DPPH free radicals were the most outstanding at any test concentration, and GANC containing three hydroxyl groups had a stronger ability to scavenge DPPH free radicals than PANC containing two hydroxyl groups. Specifically, the scavenging rate of GANC was 99.89 ± 0.02%, while PANC was 91.72 ± 0.10%. Finally, for GANC and PANC, the scavenging effect against DPPH free radicals at the highest concentration still showed a relatively strong upward trend. This meant that if the concentration of the samples increased, their antioxidant ability against DPPH radicals might also be enhanced accordingly. On the whole, the antioxidant activity of polyphenol-functionalized COS pyridinium salts was superior to that of the others at test concentrations, and the derivatives with a higher content of phenolic hydroxyl groups showed a more pronounced ability to scavenge DPPH free radicals compared to their counterparts.

**Figure 4 marinedrugs-23-00150-f004:**
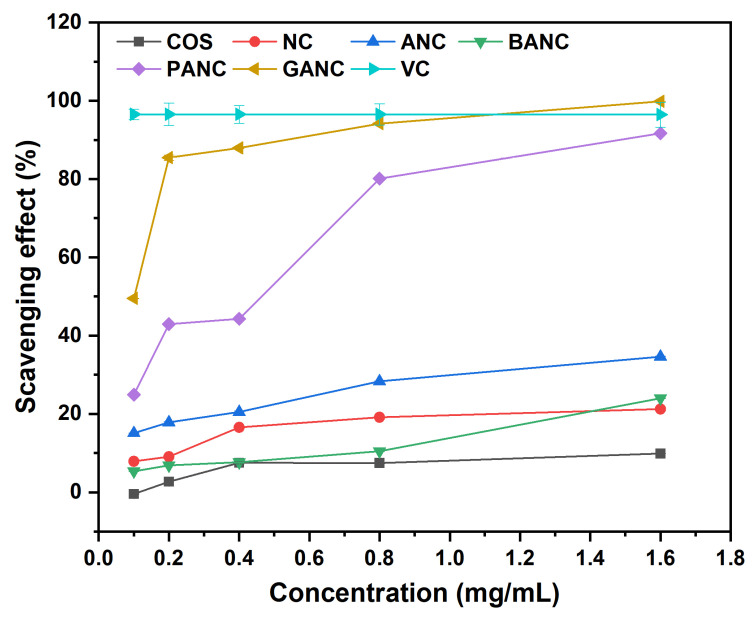
The scavenging effect against DPPH radical of COS and its derivatives. (NC, ANC, BANC, PANC, and GANC, respectively, represent nicotinated COS, aminoethyl nicotinated COS, benzoic acid-functionalized COS pyridinium salt, protocatechuic acid-functionalized COS pyridinium salt, and gallic acid-functionalized COS pyridinium salt. VC stands for ascorbic acid).

#### 2.5.2. The Scavenging Ability of the Superoxide Anion Radical

Produced in the redox pathway of superoxide, superoxide anion radicals are one of the most dangerous highly reactive oxygen species in aerobic cells. Several chronic causes including cancer, diabetes, aging, and neurodegenerative diseases could be linked to the harmful effects of superoxide anion radicals [45]. Hence, there is no doubt that it is of great significance to remove the excess superoxide anion radicals from living organisms in a timely manner [46]. In this part, superoxide anion radicals are generated by the reaction of PMS with reducing agent NADH, followed by being interacted with NBT to form the blue substance diformazan with a maximum absorption wavelength at 560 nm [44,47]. The scavenging effect of polyphenol-functionalized COS pyridinium salts against superoxide anion radicals is shown in Figure 5. It could be clearly seen from the graph that the scavenging capacity of all the tested samples increased with increasing concentration except for NC, and the clearance rate of superoxide anions radicals reached the maximum at the test concentration of 1.6 mg/mL. In particular, the scavenging values of ANC, PANC, and GANC on superoxide anions radicals at 1.60 mg/mL were 99.24 ± 0.53%, 98.90 ± 0.82%, and 99.99 ± 1.10%, respectively. Notably, NC showed a lower antioxidant effect than COS [46].

**Figure 5 marinedrugs-23-00150-f005:**
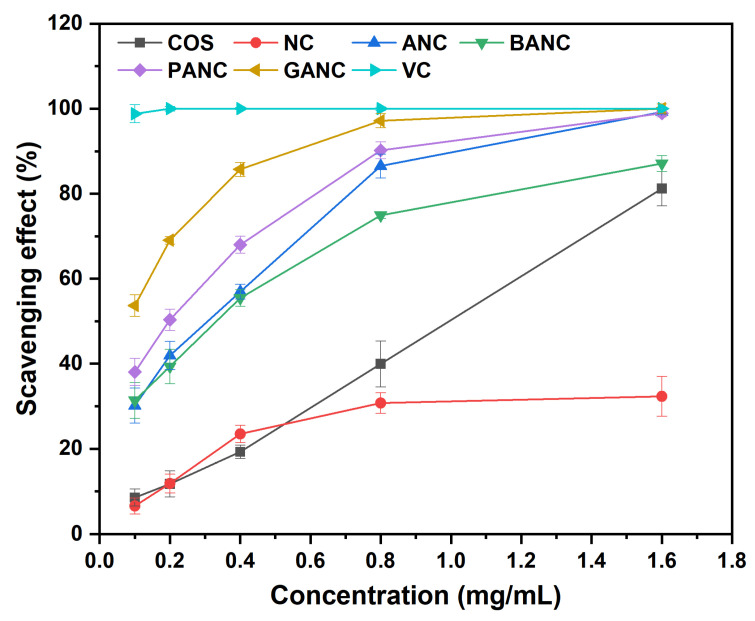
The scavenging effect against superoxide anion radical of COS and its derivatives. (NC, ANC, BANC, PANC, and GANC, respectively, represent nicotinated COS, aminoethyl nicotinated COS, benzoic acid-functionalized COS pyridinium salt, protocatechuic acid-functionalized COS pyridinium salt, and gallic acid-functionalized COS pyridinium salt. VC stands for ascorbic acid).

#### 2.5.3. Reducing Power

Reducing power is an important index in evaluating the antioxidant activity of a substance. Here, K_3_Fe(CN)_6_ can be reduced to K_4_Fe(CN)_6_ by antioxidants, and K_4_Fe(CN)_6_ is complexed with Fe^3+^ to form a dark blue precipitate Fe_4_[Fe(CN)_6_]_3_, which greatly enhances the absorbance of the reaction system at the maximum absorption wavelength of 700 nm [9]. The reducing power of polyphenol-functionalized COS pyridinium salts is shown in Figure 6. Overall, the antioxidant activity of the prepared samples increased with the increase in concentration, but the reducing power of COS and NC was in a horizontal state with almost no change. Observably, it could be seen that GANC and PANC exhibited excellent reducing power, and the antioxidant activity of the former was more prominent in the test concentration range of 0.10–1.60 mg/mL. The reducing power of COS and its derivatives was ranked at test concentrations as follows: GANC > PANC > BANC > ANC > NC > COS.

**Figure 6 marinedrugs-23-00150-f006:**
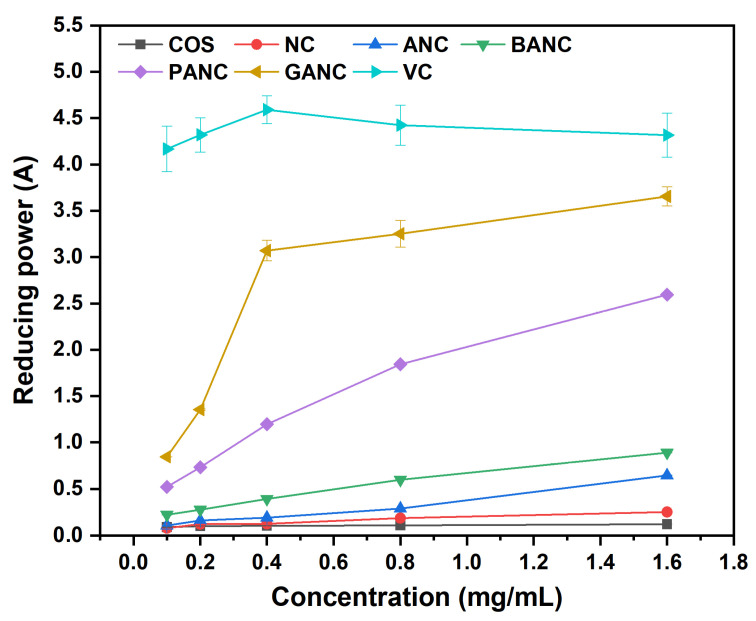
The reducing power of COS and its derivatives. (NC, ANC, BANC, PANC, and GANC, respectively, represent nicotinated COS, aminoethyl nicotinated COS, benzoic acid-functionalized COS pyridinium salt, protocatechuic acid-functionalized COS pyridinium salt, and gallic acid-functionalized COS pyridinium salt. VC stands for ascorbic acid).

### 2.6. Cytotoxicity Assay

The cytotoxicity of polyphenol-functionalized COS pyridinium salts was determined by MTT assay against L929 cells, and the test concentrations selected in this experiment were 10, 50, 100, 500, and 1000 μg/mL. The corresponding experiment results are shown in Figure 7. The cell viability of L929 cells in the presence of COS and its derivatives at tested concentrations all exceeded 100%. Moreover, the cell viability of L929 cells gradually increased with increasing concentration. Notably, PANC and GANC exhibited particularly favorable effects on the survival of L929 cells. As shown on the diagram, the cell viability of both was more than 140% at maximum experimental concentration.

## 3. Discussion

In this study, COS was used as the lead compound to synthesize several derivatives containing bioactive phenolic acids and a unique pyridinium salt structure through niacylation, salt-forming, and grafting reactions. The successful synthesis of the target compounds was confirmed by the identification of active functional groups (hydroxyl, amino, and aromatic ring) using FT-IR, and the detection of labile hydrogens through ^1^H NMR. As a marine-derived natural polymer, CS/COS possesses versatile chemical modification potential due to their active functional groups [47]. Numerous studies have focused on grafting phenolic compounds directly onto CS/COS to enhance its antioxidant activity [48]. However, as far as we know, the solubility of the antioxidants is closely related to their antioxidant activity, with water-soluble antioxidants generally exhibiting enhanced activity [49]. To ensure the water solubility of the COS-phenolic acid conjugates, we designed the introduction of a pyridinium salt structure into the chemical modification process, a strategy not previously reported in the literature for improving the water solubility of antioxidants.

To quantify the DS of the compounds under chemical modification, we performed integration using ^1^H NMR. By analyzing the area of the corresponding signals in the ^1^H NMR spectrum, we were able to determine the extent of the combination between the two compounds [50]. The DS of NC prepared via CDI catalysis indicated that CDI was an excellent catalyst for the formation of ester and amide bonds, consistent with the report by Lin et al. [40]. In contrast, the DS of the three target compounds prepared via EDCI/NHS coupling was relatively low, possibly because the chemical reagents EDCI/NHS were not conducive to the combination of amino and carboxyl groups. However, the differences in the DS among the three compounds were not significant, allowing us to rule out the possibility that differences in DS caused the differences in antioxidant activity, that is, the differences in antioxidant activity were solely related to the structure of the compounds.

Solubility testing was an important part of this study. We dissolved COS and its derivatives in equal amounts in six common solvents and observed significant differences. Taking aqueous solution as an example, NC precipitated, while the other five samples exhibited good solubility [51]. As is well known, COS has excellent water solubility due to the presence of hydrophilic groups (hydroxyl and amino) and the low molecular weight. ANC, with its hydrophilic groups (amino) and pyridinium salt structure, greatly enhanced its water solubility. The transparent solutions of the target products (BANC, PANC, and GANC) also demonstrated that they had good water solubility, and the pyridinium salt structure played a significant role in enhancing the solubility of the products.

To evaluate the antioxidant activity of COS and its derivatives, we conducted DPPH radical scavenging assay, superoxide anion radical scavenging assay, and reducing power assay, using VC as the positive control to demonstrate the effectiveness of the products’ antioxidant activity. At the tested concentrations, GANC exhibited stronger antioxidant capacity than PANC, followed by BANC, which lacks phenolic hydroxyl group(s) in its structure. After analysis, the differences were highly significant (*p* < 0.05). Since the only structural difference among the three compounds is the number of phenolic hydroxyl groups, we speculated that the number of phenolic hydroxyl groups was the main reason for the differences in antioxidant activity, consistent with the study by Li et al. [37]. Moreover, as the sample concentration increased, the antioxidant activity of the products also gradually increased, showing a concentration-dependent characteristic. NC exhibited relatively low radical scavenging ability, which may be attributed to its poor water solubility [46]. In addition, comparing the antioxidant activity of VC with that of PANC and GANC, we clearly found that PANC and GANC still showed an increasing trend in antioxidant capacity at the highest concentration of 1.6 mg/mL in this experiment, suggesting that these two substances may have antioxidant activity comparable to VC at high concentrations [52].

Cytotoxicity assay is an important aspect of testing the biocompatibility of biomaterials. From the analysis results, the survival rate of L929 cells in the presence of all samples was greater than 100% at the tested concentrations, indicating that all the samples prepared were non-toxic to normal cells and could be considered to have good biocompatibility at the cellular level [46]. Furthermore, as the sample concentration increased, the survival rate of L929 cells also showed an upward trend, that is, the higher the sample concentration, the higher the cell survival rate. Among all the tested samples, PANC and GANC had a significant impact on the survival rate of L929 cells, which may be attributed to their excellent antioxidant activity. Studies have reported that compounds with strong antioxidant activity can enhance the interaction with intracellular ROS, thereby increasing cell survival rates and promoting cell proliferation [37].

In summary, this study synthesized compounds containing bioactive phenolic acids and pyridinium salts through three steps. Through solubility analysis, antioxidant activity testing, and cytotoxicity assay, the target products were shown to have good water solubility, enhanced antioxidant activity, and good biocompatibility. The products prepared in this study, with their excellent characteristics, could be used as natural antioxidants in the food and pharmaceutical industries.

## 4. Materials and Methods

### 4.1. Materials

Chitooligosaccharide lactate with 95% *N*-deacetylation and a molecular weight of less than 3 kDa was purchased from Shandong Weikang Biomedical Technology Co., Ltd. (Linyi, China). Nicotinic acid, *N*,*N*’-carbonyl diimidazole (CDI), *2*-bromoethylamine hydrobromide, ascorbic acid (VC), protocatechuic acid, and gallic acid were provided by Macklin Biochemical Co., Ltd. (Shanghai, China). Benzoic acid, dimethyl sulfoxide (DMSO), *N*,*N*’-dimethylformamide (DMF), *N*-methyl-*2*-pyrrolidone (NMP), anhydrous ethanol, and acetone were supplied by Sinopharm Chemical Reagent Co., Ltd. (Shanghai, China). All the above chemical reagents were analytical grade and used as received without further purification. L929 cells (GDC0034) were obtained from the China Center for Type Culture Collection.

### 4.2. Chemical Synthesis

#### 4.2.1. Synthesis of Nicotinated Chitooligosaccharide (NC)

NC was prepared according to the following procedure: nicotinic acid (2.46 g, 20 mmol) was dispersed in a 50 mL round-bottomed reaction flask containing 5 mL of DMSO and well stirred for 30 min. Then an equimolar amount of CDI (3.24 g) was distributed in it slowly to further react for 12 h at 60 °C under an N_2_ atmosphere. Whereafter, the activated niacin solution was added into another flask containing a solution of 1.61 g of COS lactate dissolving in 5 mL of DMSO at room temperature. Then the mixture was reacted with a vigorous stir at 60 °C under the protection of N_2_. After 12 h of reaction, the mixture system was cooled down to 25 °C and the NC product was precipitated and washed three times by anhydrous ethanol ahead of freeze-drying at −50 °C for 24 h.

#### 4.2.2. Synthesis of Aminoethyl Nicotinated Chitooligosaccharide (ANC)

To prepare the desired compound ANC, NC (5.30 g) was dissolved in 30 mL of DMSO in a round-bottomed reaction flask, followed by adding *2*-bromoethylamine hydrobromide (12.30 g, 60 mmol), and stirring well to obtain a clear solution. The mixture system was reacted for 24 h at 60 °C under an N_2_ atmosphere. Subsequently, the reaction mixture was cooled down to 25 °C and then poured into anhydrous ethanol to precipitate and rinse three times. The desired product (ANC) was finally obtained after vacuum freeze-drying for 24 h.

#### 4.2.3. Synthesis of Benzoic Acid, Protocatechuic Acid, and Gallic Acid-Functionalized Chitooligosaccharide Pyridinium Salts (BANC, PANC, and GANC)

The synthesis of BANC, PANC, and GANC was performed by the EDCI/NHS coupling reaction. Specifically, 9 mmol of benzoic acid (1.10 g), protocatechuic acid (1.39 g), and gallic acid (1.53 g) were respectively dissolved in 5 mL DMF, then three portions of EDCI (9 mmol) were separately added to the above solution with a good stir until the solutions became clear. The mixture solutions were constantly reacted for 0.5 h at 25 °C. Subsequently, NHS (9 mmol) was respectively put into the above reaction solutions under an ice bath and continued to react for 1 h. ANC (1.17 g) was dissolved in 5 mL of DMSO. Whereafter, the activated reaction solutions obtained in the first step were added in drops to the solution of ANC apart. Then three reaction systems protected by N_2_ were reacted at 25 °C. After 24 h of reaction, the mixture reactions were precipitated and washed three times with anhydrous ethanol. Three products of BANC, PANC, and GANC were collected after vacuum freeze-drying for 24 h.

### 4.3. Analytical Methods

#### 4.3.1. Fourier Transform Infrared (FT-IR) Spectroscopy

The FT-IR spectra of polyphenol-functionalized COS pyridinium salts were carried out by using Nicolet iS50 Fourier Transform Infrared Spectrometer (Thermo, Waltham, MA, USA) at a resolution of 4.0 cm^−1^ in the scope of 4000–400 cm^−1^ range. The sample was freeze-dried, and analytical grade KBr was thoroughly mixed, ground into powder, pressed into tablet form at 25 °C, and scanned 16 times.

#### 4.3.2. Proton Nuclear Magnetic Resonance (^1^H NMR) Spectroscopy

The ^1^H NMR spectra of polyphenol-functionalized COS pyridinium salts were used to further investigate their chemical structures by Bruker AVIII-500 Spectrometer purchased from Bruker Tech. and Serv. Co., Ltd. (500 MHz, Switzerland, Beijing, China) at 25 °C. Except for NC, all tested samples (20.00 mg) were dissolved in 0.60 mL of deuterium oxide (D_2_O) for analysis. Due to the poor solubility of NC in water, in addition to adding 0.60 mL of D_2_O, an additional 5 μL of concentrated hydrochloric acid was also added to facilitate its dissolution.

#### 4.3.3. Degrees of Substitution (DS)

The DS of COS and its derivatives were quantitatively determined by hydrogen signals obtained through ^1^H NMR spectra, and calculated by the following Equations:(1)$${\mathrm{DS}}_{1}=\frac{6\int H_{a-d}}{4\int H_{2-6}}$$(2)$${\mathrm{DS}}_{2}=\frac{4\int H_{e}}{2\int H_{a-d}}\times {\mathrm{DS}}_{1}$$(3)$${\mathrm{DS}}_{3}=\frac{2\left(\int H_{a-g}-\int H_{a-d}\right)}{n\int H_{e}}\times {\mathrm{DS}}_{2}$$
where DS_1_ represents the DS of nicotinoyl onto the COS backbone, ∫H_2–6_ is the integral values of the hydrogen atoms bonded to C2–C6 of natural COS backbone (2.49–4.60 ppm), ∫H_a–d_ is the integral values of the hydrogen atoms on the pyridine ring (7.70–9.31 ppm). DS_2_ indicates the DS of aminoethyl grafted onto COS pyridinium salts, ∫He is the integral values of the hydrogen atoms on methylene that are relatively close to the nitrogen cation. DS_3_ shows the DS of aromatic rings grafted onto ANC, ∫H_a–g_ is the integral values of the hydrogen atoms on benzoic acid, protocatechuic acid, and gallic acid, respectively (located at 6.48–10.00 ppm, 6.50–10.00 ppm, and 6.94–9.73 ppm, severally), and the corresponding *n* is equal to 5, 3, and 2.

### 4.4. Solubility Assay

In order to test the solubility of the prepared compounds in common solvents, the simple experimental procedure was as follows: A sample of 10.00 mg was weighed into the test tube. Subsequently, 1 mL of solvents (deionized water, dimethyl sulfoxide, *N*,*N*’-dimethylformamide, *N*-methyl-*2*-pyrrolidone, anhydrous ethanol, and acetone) were respectively added and shaken well at room temperature for 10 min, and the dissolution effects of the samples were visually observed.

### 4.5. Antioxidant Assays In Vitro

#### 4.5.1. DPPH Radical Scavenging Ability Assay

The scavenging activities of polyphenol-functionalized COS pyridinium salts against DPPH radical were determined in accordance with a previous method with slight alteration [41]. Firstly, the sample was prepared at a concentration of 10 mg/mL. Afterward, 1 mL of DPPH-ethanol (180.0 μmol/L) solution was added to 0.5 mL of sample solution with different concentrations in the test tube to make the final solution concentrations of 0.10, 0.20, 0.40, 0.80, and 1.60 mg/mL, respectively. Meanwhile, 0.5 mL of sample solution was substituted with deionized water as a blank group, and 1.0 mL of DPPH-ethanol solution was displaced with absolute ethanol as the control group. VC was used as the positive control. Whereafter, the mixture reaction system was shaken evenly and incubated for 20 min in complete protection from light. Then, the absorbance of all solutions was tested at 517 nm using a UV-visible spectrophotometer. The DPPH radical scavenging ability of the above samples was calculated according to the following formula:(4)$$\mathrm{Scavenging}\;\mathrm{effect}\;(\%)=\left[1-\frac{A_{\mathrm{sample}}-A_{\mathrm{control}}}{A_{\mathrm{blank}}}\right]\times 100$$
where A_sample_, A_control_, and A_blank_ represent the absorbance of the samples, the control, and the blank at 517 nm, respectively.

#### 4.5.2. Superoxide Anion Radical Scavenging Activity Assay

The superoxide anion radical scavenging abilities of polyphenol-functionalized COS pyridinium salts were assessed according to the model of Tan’s method [53]. Firstly, the sample was prepared at a concentration of 10 mg/mL. Afterward, 0.5 mL of nicotinamide adenine dinucleotide reduced (NADH, 338 μM), 0.5 mL of nitro blue tetrazolium (NBT, 72 μM), and 0.5 mL of phenazine methosulfate (PMS, 30 μM) in Tris-HCl buffer (16 mM, pH = 8.0) were successively added to 1.5 mL of the sample solution in the test tube to make the final solution concentrations of 0.10, 0.20, 0.40, 0.80, and 1.60 mg/mL, severally. In addition, all the sample solutions were replaced by 1.5 mL of deionized water as a blank group and 0.5 mL of NADH was substituted with equivalent Tris-HCl as the control group. VC was used as the positive control. Then, the reaction solution was shaken evenly and incubated at room temperature in the dark for 5 min. Subsequently, the absorbance of all solutions was measured quickly at 560 nm. The superoxide anion radical scavenging ability of the above samples was calculated according to the following formula:(5)$$\mathrm{Scavenging}\;\mathrm{effect}\;(\%)=\left[1-\frac{A_{\mathrm{sample}}-A_{\mathrm{control}}}{A_{\mathrm{blank}}}\right]\times 100$$
where A_sample_, A_control_, and A_blank_ represent the absorbance of the samples, the control, and the blank at 560 nm, respectively.

#### 4.5.3. Reducing Power Assay

The reducing powers of polyphenol-functionalized COS pyridinium salts were measured by the procedure reported previously with a slight modification [54]. Firstly, the sample was prepared at a concentration of 10 mg/mL. Afterward, 0.5 mL of potassium ferricyanide (1%, *w*/*v*) solution in sodium phosphate buffer (0.2 M, pH = 6.6) was added to 0.5 mL of sample solution. Then the resulting solution was shaken evenly and incubated at 50 °C for 20 min. After adding 0.5 mL of trichloroacetic acid (10%, *w*/*v*) to the mixture at 25 °C, the solution was shaken well and successively centrifuged at 3000 rpm for 5 min. Whereafter, 0.75 mL of the upper layer was blended with distilled water (0.6 mL) and ferric chloride (0.15 mL, 0.1%, *w*/*v*) without light for 10 min, to obtain the end concentrations of 0.10, 0.20, 0.40, 0.80, and 1.60 mg/mL. VC was used as the positive control. Finally, the absorbance of all solutions was carried out at 700 nm.

### 4.6. Cytotoxicity Assay

The cytotoxicity of polyphenol-functionalized COS pyridinium salts against L929 cells was determined by MTT assay [55]. Firstly, L929 cells were cultured in DMEM medium (including 10% fetal bovine serum and 1% penicillin/streptomycin) at 37 °C under an atmosphere of 5% CO_2_. Then, the cells were transferred and seeded on 96-well flat-bottom culture plates at a density of 1.0 × 10^5^ cells and incubated with the same environmental conditions as cultured cells for 12 h. Whereafter, the supernatant was removed and the cells were cultured in a DMEM medium containing different concentrations (10, 50, 100, 500, and 1000 μg/mL) of COS and its derivatives for 24 h. Then, the medium was poured off and 100 μL of MTT solution was added to each well. After 4 h of incubation at 37 °C, MTT solution was replaced with 150 μL of DMSO. The absorbance of each well at 490 nm was measured by the microplate reader. In this part, the blank group only contained DMEM medium and MTT solution, and a mixture of cells and MTT solution served as a negative control. The cytotoxicity assay of the above samples was calculated according to the following formula:(6)$$\mathrm{Cell}\;\mathrm{viability}\;(\%)=\frac{A_{\mathrm{sample}}-A_{\mathrm{blank}}}{A_{\mathrm{negative}}-A_{\mathrm{blank}}}\times 100$$
where A_sample_, A_blank_, and A_negative_ represent the absorbance of the samples, the blank, and the negative control at 490 nm, respectively.

### 4.7. Statistical Analysis

All experiments were conducted at least three times and data were presented as the mean ± standard deviation (SD). Comparisons between groups through one-way analysis of variance (ANOVA) were considered statistically significant when *p* < 0.05.

## 5. Conclusions

To improve the water solubility of COS-grafted phenolic acid derivatives, this paper successfully designed and synthesized polyphenol-functionalized COS pyridinium salts. The antioxidant activity against three kinds of free radicals, cytotoxicity against L929 cells, and solubility at common solvents were investigated. The antioxidant assays indicated that polyphenol-functionalized COS pyridinium salts had enhanced antioxidant properties, and the more the number of phenolic hydroxyl groups, the stronger the antioxidant ability. The cytotoxicity experiment result showed that the cell viability of end products prepared exceeded 100%, which suggested that these compounds might have a certain effect on promoting cell proliferation. By solubility experiment, designed target compounds exhibited good water solubility, which was attributed to the presence of pyridinium in their structures. Overall, the polyphenol-functionalized COS pyridinium salts with good biocompatibility and strong antioxidant ability could be potential and useful candidates to be developed as newly safe antioxidants for use in pharmaceutical and food industries.

## Data Availability

The data presented in this study are available on request from the corresponding author.

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
