# Peer review of "Preparation and Characterization of Polyphenol-Functionalized Chitooligosaccharide Pyridinium Salts with Antioxidant Activity"

_marinedrugs, 2025, doi:10.3390/md23040150_

Round 1
Reviewer 1 Report
Comments and Suggestions for Authors
Dear Editor,
I have reviewed the manuscript titled "Preparation and characterization of polyphenol-functionalized chitooligosaccharide pyridinium salts with antioxidant activity." The manuscript is very interesting and discusses the preparation and evaluation of novel polyphenol-functionalized COS pyridinium salts with their antioxidant and cytotoxicity activities. But the manuscript has some deficiencies and below are some queries and suggestions. The manuscript can be accepted after major revision.
- Add some literature regarding benzoic acid, protocatechuic acid, gallic acid with respect to physicochemical properties and applications in the introduction.
- Why wasn't standard reference (such as ascorbic acid) material used to check the comparative results for different activities with prepared compounds?
- Results are not discussed properly with respect to statistical analysis (p > or< 0.05, 0.01, 0.001)
- Figure 4, Figure 5 and Figure 6 need to be redrawn, they were blurring on zooming.
- Results are available, but no discussion part is available.
- What was kept as a control sample during cell cytotoxicity studies?
- Line 316: Specifically, 9 mmol of compounds (benzoic acid, protocatechuic acid and gallic acid) were respectively dissolved in 5 mL DMF, how much quantity was added of 9 mmol?
- Line 434: By solubility experiment, designed target compounds exhibited good water solubility. The authors did not perform saturated solubility analysis, then how they can say that compounds showed good water solubility?
- Check the manuscript for plagiarism, it should be below 15%.
Reviewer 2 Report
Comments and Suggestions for Authors
This MS should be improved before being accepted for further steps. Here are the points:
- The link of this study to Marine drugs’s scope is not sufficiently addressed. Thus, the authors are suggested to improve the introduction section to make this study fit the journal's scope.
- The discussion section is missing.
- The results of this study need to be discussed with previous studies, especially since studies on antioxidant activity from chitosan derivatives are relatively common.
- The antioxidant activity of COS and its derivatives should be compared with control to clarify their effectiveness further.
- The legend of figures should be improved. The COS derivatives' names and data’s information should be added to figure legends.
- Figure 2. PANS or PANC.
- Markers in Figures 4, 5,&6 are too small. Please improve them.
- Figure 4. The authors were asked to check the PANC data at 0.4 mg/mL. This data point is abnormal compared to the data trend.
- Degrees of substitution analysis should be combined with H MNR spectra.
- The reference should follow the rules of the journal.
Reviewer 3 Report
Comments and Suggestions for Authors
This paper, entitled "Preparation and characterization of polyphenol-functionalized chitooligosaccharide pyridinium salts with antioxidant activity," presents the study. Novel polyphenol-functionalized COS pyridinium salts were synthesized via niacin acylation. The chemical structures of the derivatives were characterized by FT-IR and 1H NMR. Additionally, a DPPH radical scavenging assay, a superoxide radical scavenging assay, and a reducing power assay were used. The solubility of synthesized derivatives in several common solvents was also analyzed. The cytotoxicity experiment of new polyphenol-functionalized COS pyridinium salts was also performed.
This work does not include rigorous assays such as DSC, TGA-DTGA, and XRD, whose assays would support the conclusions from a perspective complementary to the one observed. Therefore, the authors must complete these assays for the paper to be acceptable for publication.
Comments on the Quality of English Language
No comments.
Round 2
Reviewer 2 Report
Comments and Suggestions for Authors
The manuscript has been revised.
Reviewer 3 Report
Comments and Suggestions for Authors
The paper must be accepted for publication.